# The MATES Case Management Model: Presenting Problems and Referral Pathways for a Novel Peer-Led Approach to Addressing Suicide in the Construction Industry

**DOI:** 10.3390/ijerph18136740

**Published:** 2021-06-23

**Authors:** Christopher M. Doran, Lisa Wittenhagen, Edward Heffernan, Carla Meurk

**Affiliations:** 1Cluster for Resilience and Wellbeing, Appleton Institute, Central Queensland University, Brisbane, QLD 4000, Australia; 2Faculty of Medicine, School of Public Health, The University of Queensland, Brisbane, QLD 4072, Australia; l.wittenhagen@uq.edu.au (L.W.); Ed.Heffernan@health.qld.gov.au (E.H.); Carla.Meurk@health.qld.gov.au (C.M.); 3Queensland Centre for Mental Health Research, Wacol, QLD 4076, Australia; 4Queensland Forensic Mental Health Service, Brisbane, QLD 4000, Australia

**Keywords:** case management, help seeking behaviour, construction industry workers, suicide prevention, MATES in Construction

## Abstract

MATES in Construction (MATES) is a multimodal, peer-led, workplace suicide prevention and early intervention program developed to reduce the risk of suicide among construction industry workers through active facilitation of appropriate support. The MATES case management model provides an example of a nonclinical service for meeting the needs of individuals in the construction industry who, while at elevated risk of mental health problems and suicidality, are traditionally less likely to seek help. The aim of this research was to conduct an evaluation of the MATES case management database to quantify service demand, and to examine the demographic, occupational profile, presenting issues, referral pathways, and perceived benefit of case management among individuals who used this service. The research reports on routinely collected data from the Queensland MATES case management database, which contains records on 3759 individuals collected over the period 2010–2018, and findings from a small and opportunistic exit survey undertaken with 14 clients in 2019. Overall, findings suggest that the demand for case management through MATES has increased significantly and that clients felt that their needs and concerns were appropriately addressed. The most common presenting issues were relationship, work, and family problems, suicide, and mental health concerns. Findings confirm that causes of distress extend beyond the realm of mental disorder and span a range of psychosocial issues. Significantly, it offers an approach that may divert individuals in crisis away from presenting to over-run emergency departments, and towards services that are more equipped to meet their individual needs.

## 1. Introduction

Suicide rates are generally higher in men, with men who work in the construction industry at a particularly elevated risk [1,2,3,4]. In Australia, lower-skilled trade-workers have an adjusted suicide rate of 18 per 100,000, greater than that of higher-skilled trade-workers, who have an adjusted rate of 13 per 100,000, and that of the general male population^3^. Reviews of international studies have found that labourers and cleaners have an overall increased risk of suicide, relative to the general working population, machine operators, and agricultural workers [4]. Risk factors for suicide among construction industry workers include mental health problems, employment instability, workplace injury or work limiting illness, financial or legal problems, relationship breakdowns, disputes over child custody, and substance use^2^. At the same time, and perhaps contributing to this elevated risk, men are less likely to seek help for mental health problems or suicidality [5,6,7].

Mates in Construction (MATES) is an example of a multifaceted strategy developed in Australia to address suicide prevention in the workplace. MATES was established in 2008 by the Building Employees Redundancy Trust to prevent suicide in the construction industry [8]. It is a multimodal, nonclinical, industry-led, peer-based workplace suicide prevention and early intervention program, consistent with the Living Is for Everyone (LIFE) strategy [9] and Mrazek and Haggerty’s spectrum of prevention and intervention [10]. MATES provides a range of mental-health-related training, offers nonclinical case management, an outreach service, and a twenty-four-hour support service to employees of the construction industry.

Since its inception, MATES has had substantial uptake in the building and construction sector and has developed an evidence-base supporting its effectiveness. Previous evaluation research has demonstrated the social validity of the program among construction workers [11], effectiveness in shifting beliefs around suicide [12,13], improvements in suicide prevention literacy, and increased intentions among workers to offer help to workmates and to seek help for themselves [13,14,15]. Research has also demonstrated the significant economic return of investing in workplace suicide prevention initiatives such as MATES [16,17,18]. No study has examined the potential impact of the MATES approach to case management.

Case management models for suicide prevention vary in numerous ways, including referral pathways to a service (many are via emergency departments or other health services); whether case managers provide any psychological treatment or behavioural activation; the role of mental health professionals (versus nonmental health professionals) in case management and/or reviews of case managed clients; frequency and duration of contact (between 4 weeks and 18 months, with varying patterns of follow-up noted); differences in modalities of follow-up (face-to-face, telephone, home visits or a combination of one or more of these); whether or not clinical suicide risk assessment or formal monitoring is carried out; and the role of safety planning. There is limited evidence regarding the efficacy of case management for suicidal individuals [19,20,21,22,23]. Published studies over the past five years tend to focus on individuals who present to emergency departments as a result of a suicide attempt and are therefore unlikely to be generalisable to prevention or early intervention services, such as MATES. Findings from these studies are mixed, but indicate that case management models are associated with modest reductions in suicide risk [21,22,24,25,26]. Only one study published in 2007 examined a nonclinical case management approach in patients discharged from psychiatric inpatient settings [19]. De Leo and Heller (2007) reported that people randomised to receive intensive case management displayed significant improvements in depression scores, suicide ideation, and quality of life, compared with those in the usual treatment group [19].

MATES case managers do not provide mental health care to clients. Rather, MATES utilises the brokerage model—a brief approach to case management in which case managers attempt to help clients to identify their needs and broker supportive services over a brief contact period [11]. This model assumes that a client in need will voluntarily use the services once they know that they are available and learn how to access them. This model works best when a client’s biggest challenge is access to services, rather than availability of services. In a brokerage case management model, the case manager/social worker provides very little direct service to the client. Instead, they serve as a link between a client and community resources. The focus is on assessing needs, planning a service strategy, and connecting and following up with clients [27].

The aim of this research was to conduct an evaluation of the MATES case management database to quantify service demand, and to examine the demographic, occupational profile, presenting issues, referral pathways, and perceived benefit of case management among individuals who used this service.

## 2. Materials and Methods

The evaluation relied on two data sources: (1) routinely collected data held in the Queensland MATES case management database spanning the period January 2010–December 2018; and (2) data from an exit survey administered to case management clients during June 2019. Data held in the MATES case management database allowed us to use naturally occurring, routinely collected data to reliably assess characteristics of service use, including the demographic characteristics of the individuals making use of the service. A self-report survey was used to obtain information about the perceived benefit of case management among individuals who used the services provided by MATES.

### 2.1. Routinely Collected Administrative Data

The Queensland MATES case management database contains sociodemographic information including the age, gender, and occupation of individuals who contacted MATES, information about how individuals were connected with MATES, what their presenting concerns were, and which services they were referred to. In addition, the MATES case manager records whether an individual was assessed as being a suicide risk and the number of case notes that were recorded for each MATES client.

For the purpose of analysis, occupations were grouped based on the Australian Bureau of Statistics and New Zealand Standard Classifications of Occupations [28]. All presenting concerns listed within the MATES case management database were grouped into 16 categories (Accommodation, Alcohol, Anger, Drugs, Family, Fatality or Grief, Financial, Gambling, Health or Injury, Inquiry, Legal, Mental Health, Past Issues, Relationship, Suicide-Related, Work-Related). Options for referral pathways linking clients with the MATES case management service were grouped into six categories (Family, MATES services, Other Service/Training Provide, Self-Referred, Union, Other). Services that clients could be referred to following a contact with MATES were grouped into 11 categories (Alcohol and Drug Services, Ambulance or Police, an Employee Assistance Program (EAP) that provides consulting and training with the aim of creating mentally healthy workplaces, Employment and Training Services, Housing Financial and Human Services, Insurance Scheme, Legal Services, Medical Professional, Mental Health, Counselling, or Wellbeing Services, No Referral Required, Other). Case notes refer to a chronological record of interactions, observations, and actions relating to a client.

### 2.2. Exit Survey

For the purpose of this study, a brief exit survey was developed by MATES case managers and the research team. Specifically, the aim of the survey was to obtain feedback from MATES clients to canvass the potential benefits of case management. The content of the survey reflected the role of the case managers in supporting clients and consisted of five closed client feedback questions and one open-ended question (Appendix A). The closed-ended questions included whether the client felt that the nature of their concerns was addressed during the case management process; whether the services were appropriate in meeting needs; whether the clients felt actively involved in the decision-making process; whether their medical, emotional, mental well-being, and spiritual needs were addressed; and whether they would recommend MATES to co-workers, family, and friends. Responses to the closed questions were recorded using a 5-item Likert scale (absolutely yes; yes; maybe; no; absolutely no). Clients’ comments around MATES in general or their case management experience (open-ended question 6) were recorded as text. The survey was administered opportunistically during telephone contact between case managers and a sample of MATES clients at the time their cases were closed in June 2019. No data were recorded on clients that were approached and/or declined.

### 2.3. Analysis

Data regarding the number of clients, records in the MATES case management database, gender, age, occupation, presenting concerns, and referral pathways to and from MATES were analysed descriptively (*n* and %). A chi-square test of independence was performed to examine the relation between the service providers MATES clients were referred to and the presenting concerns of these clients. A Wilcoxon–Mann–Whitney test was employed to assess potential associations between the number of case notes and suicidality, using the wilcox_test and wilcox_effsize functions embedded within the rstatix R package [29]. The unit of analysis was case records (i.e., the number of single contacts recorded in the MATES case management database). Data were examined for trends over time, by calculating the percentage of change (increase and decrease). Exit survey data were analysed descriptively and qualitatively. Data were analysed using *R* [30].

## 3. Results

### 3.1. Demographic Characteristics

Figure 1 summarises the change in the records over the study period. Between January 2010 and December 2018 (inclusive), the Queensland MATES case management database had 4220 records linked to 3759 individuals, of which 461 were duplicate records linked to 364 unique clients (Table 1). From 2010 to 2018, there was a 265% increase in case management records, with a marked peak in 2016 (728 case management records pertaining to 694 individuals). Trend analysis confirmed that this upward trend was significant (tau = 0.583, *p* < 0.0001). The majority of clients were male (92%), and the age of clients remained relatively stable over the study period, with a median age of 39 years (minimum age: 15 years, maximum age: 76 years). The proportion of female clients presenting to MATES for case management was highest in the years 2014 and 2016, 10% in both cases.

### 3.2. Occupation

There were 1720 records (41%) for which the occupation was either listed as ‘other’ or ‘unknown’. For records where the occupation category could be determined (*n* = 2500), the most common occupational group were labourers (30%, *n* = 757), followed by plant operators (17%, *n* = 426) and plumbers (14%, *n* = 354; Figure 2).

### 3.3. Presenting Concerns

Figure 3 shows the most common presenting concerns. Out of all records (*n* = 4220), the most common presenting concern was relationship issues (38%, *n* = 1600). This was followed by work-related concerns (27%, *n* = 1131), and family concerns (22%, *n* = 949). Suicide-related concerns (suicide ideation 11%, suicide intervention 2%, suicide bereavement 0.5%) were the fourth most frequently identified presenting issue (14%, *n* = 572). Mental health concerns were identified among 11% (*n* = 462) of all cases, and alcohol and drugs in 7% (*n* = 310) and 6% (*n* = 266) of all cases, respectively. Some changes in the prominence of presenting issues were noticeable over time. For instance, there was a noticeable increase in ‘mental health’ and ‘family’ as a presenting concern in 2018, noticeable increases in ‘suicide-related’ presentations in 2017 and 2018, and an increase in ‘work-related’ presentations in 2017. The prominence of ‘relationship’ as a presenting concern fluctuated across the nine-year period examined, between 32% in 2010 and 46% in 2011 and 2018. These changes are summarised in Figure 4.

### 3.4. Pathways to MATES Case Management Services

Out of all six possible referral categories, the most common referral pathways listed were ‘Self Referred’ (*n* = 937) and ‘MATES services’ (*n* = 936; listed for 22% of all records each), followed by referrals through a ‘Union’ (20%, *n* = 858). The other three referral source categories accounted for the remaining third of referrals to MATES case management services.

### 3.5. Services Clients Were Referred to by MATES

The provider that MATES most commonly referred their clients to was an EAP (48%, *n* = 2041). Referrals to a Mental Health, Counselling, or Wellbeing Service were listed for 12% (*n* = 526) of all records, and in about 5% (*n* = 206) of all records, a referral to a medical professional was noted. For about 25% (*n* = 1056) of all records, clients were not referred to an external service provider.

### 3.6. Number of Case Notes

The median number of case notes per record (i.e., the number of contacts with a MATES case manager) was four (IQR: 3–7). The median number of case notes for records of clients who were assessed by MATES case managers to be at risk of suicide was five (IQR: 3–8), which was significantly higher than the median number of case notes for clients that were not assessed to be at risk of suicide (*Mdn* = 4, IQR: 3–7; *Z* = 2.59, *p* < 0.01, r = 0.04).

### 3.7. Exit Surveys

The small and opportunistic online exit survey was administered via telephone by case managers to 14 clients during June 2019, before the clients’ cases were closed. Zero responses were recorded in the ‘no’ or ‘absolutely not’ category, with all 14 respondents agreeing that the nature of their concerns was addressed during the case management process (question 1) and that they would recommend MATES to co-workers, family, and friends (question 5). In all, 13 of the 14 survey participants agreed that they felt actively involved in the decision-making process (question 3) and considered that their medical, emotional, mental well-being, and spiritual needs were addressed (question 4). Twelve agreed that MATES services were appropriate in meeting client needs (question 5). Qualitative comments provided by clients to case managers identified that clients appreciated case managers ‘checking in’ with them, providing hope, and connecting them with other services. Various participants stating the following:


*“Mates was great at keeping in contact and checking in with me to make sure I was ok, while I was going up and down with my emotions and situation and making sure I stayed positive and could see a way forward.”*

*“Great support and engagement by mates, keeping me honest, checking in weekly around mental wellbeing due to my separation. Connected me into counselling and separation support services that really helped lower my mental strain and suicidal thoughts.”*

*“So thankful for the support …. as well as contacting me lots to check in and make sure that I was going ok.”*

*“Mates was really easy to engage and supported me to move through separation as well as checking in with me consistently to make sure I was ok.”*


## 4. Discussion

The role of the MATES case manager is considered key to enhancing engagement of workers with external services, providing a ‘safety net’ between the individual in need and the agencies that can help them to ensure continuity of care, ensuring that workers’ needs are being met, and advocating for further assistance where necessary [31]. MATES differs from typical case management services in that case managers do not provide direct mental health clinical care (e.g., psychological interventions) or undertake risk assessment or monitor clinical or risk variables. Rather, trained case managers work with individuals to develop a plan to address their social and care needs.

Data held in the MATES case management database allowed us to use naturally occurring, routinely collected data to reliably assess characteristics of service use, including the demographic characteristics of individuals making use of the service. A self-report survey was used to obtain information about the perceived benefit of case management among individuals who used the services provided by MATES. The reported findings from the descriptive analysis and opportunistic exit survey paint a positive picture of MATES and reinforce the important role case managers play in managing psychosocial distress in the construction industry and brokering appropriate service access.

Several findings from the descriptive analysis are noteworthy. First, the demand for case management services offered by MATES has increased markedly over time. This increased demand may be the result of many factors including increased penetration of MATES in the construction industry, together with an increased awareness of mental health issues in the workplace and society. Although these data suggest a slight decrease in case records from 2017 to 2018, MATES has experienced an overall 265% growth in case management records since 2010. Second, the majority of clients from identified occupations include labourers and operators, consistent with previous evidence of an elevated suicide risk within these occupational groups [3,4]. Third, clients tend to present for several issues, including relationship and work-related concerns. However, presenting concerns appear to have varied over time with a noticeable spike in suicide-related presentations in 2017 and 2018, and mental health presentations in 2018. These spikes may be the result of changes in coding or they could be a reflection of an increased awareness of mental health problems, though further work is required to clarify this. Fourth, the three most common referral pathways to MATES were self-referrals, referrals via MATES services, or referrals through a Union. The finding that about 22% of clients were referred to MATES case management by another MATES service reinforces the value of MATES training and the important role MATES is playing within the workforce.

Qualitative responses to the opportunistic exit survey highlighted that clients particularly valued MATES case managers checking in with clients, being generally supportive, and giving them hope during a period of distress. Clients agreed that MATES services were appropriate in meeting their needs and concerns, and that they would recommend the service to co-workers, family, and friends. Clients also appreciated the pro-active nature of case managers in checking in and making sure clients stayed positive and focussed on finding a positive way forward.

While further research is required to determine the effectiveness of the MATES case management model, the data presented here are promising and point to the acceptability of this approach for a population group known to be at elevated risk of suicide and reluctant to seek help [6,7]. For instance, a better understanding of clients (through additional data) and a better understanding of service providers (through better training or educational sessions provided by service providers) may represent a solution to better matching clients with the appropriate care required. Further, demonstrating the acceptability of a nonclinical, peer-based case management model, which is nested within a referral pathway that does not require health professional gatekeeping, represents a major advance in crisis care case management. In Australia, where this research is based, emergency departments are experiencing an inundation of mental health presentations, and individuals suffering mental health and suicide crises are more likely than those with physical health complaints to experience unduly long waiting times for treatment and admission [32,33,34,35,36,37]. There is a growing consensus, among both medical professionals and individuals with lived experience of mental health and suicide crises, that both emergency departments and mainstream mental health services are frequently not the most appropriate pathway for individuals whose distress is related to psychosocial factors, rather than specifically to mental disorder [32,33,34,35,36,37]. The MATES model, if shown to be effective, offers new directions for service provision based on care pathways with entry points and routes that are separate from, but complementary to, the health system. Given that the objective of MATES is to reduce the high level of suicide among Australian construction workers, it is timely to consider the impact that case management (and indeed, all MATES programs) may have on a client’s quality of life, workplace safety, and economic benefits.

## 5. Limitations

Some limitations are worth noting. First, the subjective nature of information collected in the case management database has not been validated. Age, occupation, presenting concerns, and referral source are all self-reported by the client. Second, although routinely collected, naturally occurring data are an excellent means of understanding service use, there is potential room for error. We found that, in general, the quality of data held for the period 5 January 2010 to 20 December 2018 in the MATES case management database was high, with about 1.72% of missing data entries (800 of 46,420 cells that were relevant for analysis). These missing records pertained to date of birth information which was either not disclosed by the client or entered in a format from which the age of a client could not be reliably retrieved. Although the majority of clients had multiple contacts with case managers, enabling validation of certain variables, there is potential room for error. Information when a client entered/left case management, number of contacts, and the providers they were referred to are considered very reliable. However, the lack of follow-up from case managers after closing a case suggests that case managers do not know if clients engaged with the referred service providers. Third, the opportunistic nature and small response to the exit survey is another limitation to this study. Given that the survey was administered to only 14 MATES clients with no identifiable information in terms of age, gender, or extent of involvement with MATES, the generalisability of responses to the entire MATES service is questionable and may not reflect the perceptions of other MATES clients. Further, although it is assumed that clients answered the questions honestly, they may have been subject to response bias as the survey was administered by case managers directly to clients who may have, therefore, felt obliged to provide a positive response, thus impacting the representativeness and generalisability of findings. As such, findings from the exit survey should be considered as potentially illustrative with a larger exit survey required to address the issue of generalisability.

## 6. Conclusions

This research contributes to the growing evidence base for the effectiveness of MATES in addressing suicide prevention in the workplace. As a multifaceted strategy introduced in 2008, MATES has enjoyed substantial uptake in the building and construction sector. The MATES philosophy of looking out for colleagues is shifting beliefs around suicide [12,13], improving suicide prevention literacy, and increasing intentions among workers to offer help to workmates and to seek help themselves [13,14,15]. This research supports the MATES case management model as a novel no-clinical service for meeting the needs of individuals in the construction industry who, while at elevated risk of mental health problems and suicidality, are traditionally less likely to seek help. Findings from this evaluation confirm that causes of distress extend beyond the realm of mental disorder and span a range of psychosocial issues. Significantly, the MATES model of case management offers an approach that may be of benefit to individuals in mental health crisis broadly and may also reduce the burden on public health systems in cases where more targeted interventions are appropriate. An important area of future research for MATES is that of data linkage for providing an objective assessment of how engagement with MATES prevents self-harming behaviours and generates a positive economic return to society.

## Figures and Tables

**Figure 1 ijerph-18-06740-f001:**
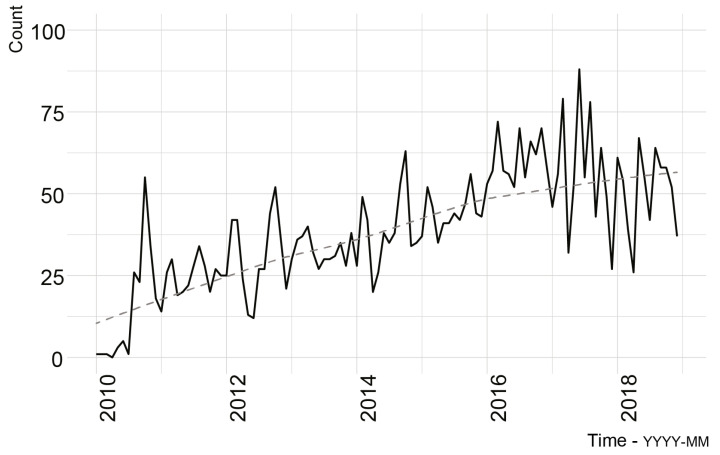
Number of records between January 2008 and December 2018 (solid) and lowest regression line (dashed).

**Figure 2 ijerph-18-06740-f002:**
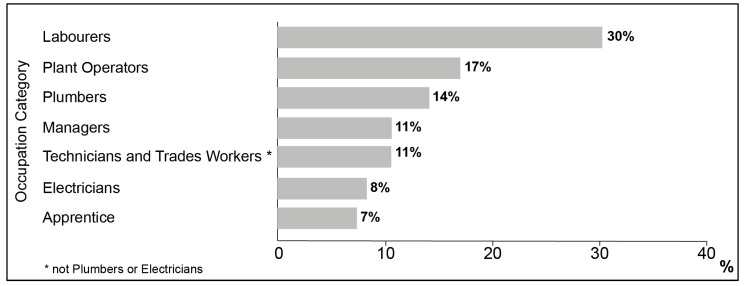
Distribution of occupation categories within the MATES case management database.

**Figure 3 ijerph-18-06740-f003:**
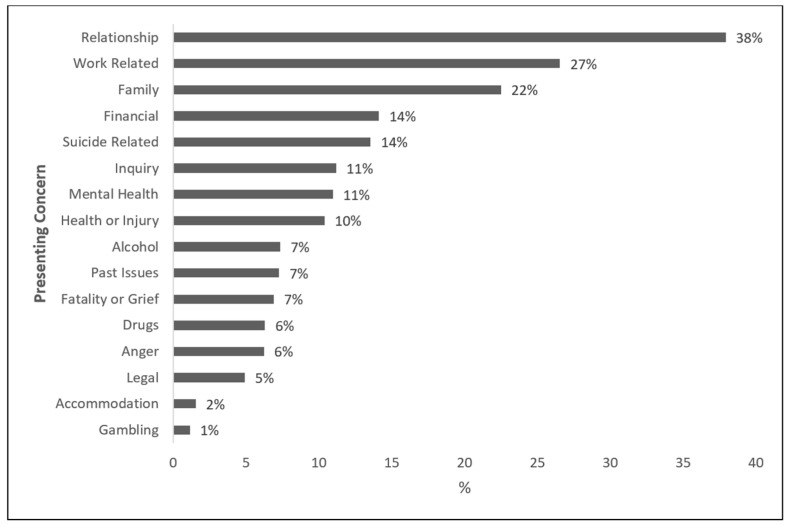
Distribution of presenting concerns within the MATES case management database.

**Figure 4 ijerph-18-06740-f004:**
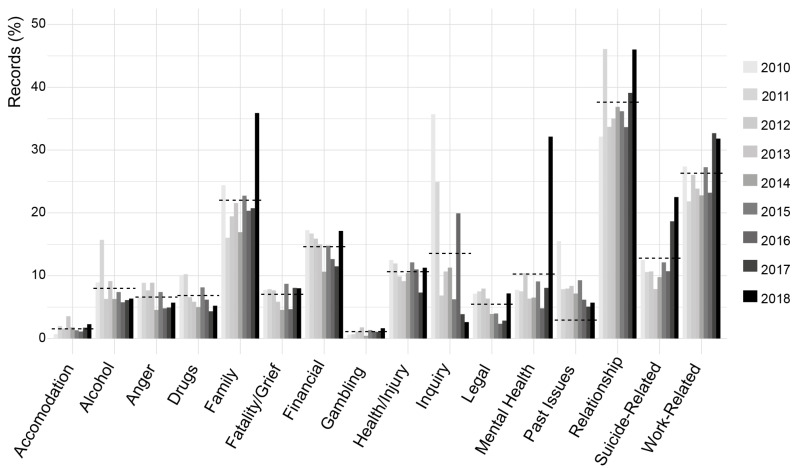
Distribution of presenting concerns over the years 2010 to 2018. Dashed lines indicate the mean percentage of records over the years 2010–2018 for each concern category.

**Table 1 ijerph-18-06740-t001:** Percentage of change, number of clients and records, gender distribution, and median age per year as recorded in the Queensland MATES case management database.

Year	Change ** Clients	N Clients	Change ** Records	N Records	Male	Female	Age Median
2010–2018 *	↑260%	3759	↑265%	4220	92%	8%	39
2018	↓10%	591	↓09%	613	92%	8%	40
2017	↓06%	654	↓08%	670	93%	7%	38
2016	↑36%	694	↑38%	728	90%	10%	39
2015	↑15%	511	↑15%	528	94%	6%	38
2014	↑16%	446	↑17%	461	90%	10%	37
2013	↑06%	384	↑08%	394	92%	8%	38
2012	↑28%	362	↑25%	365	93%	7%	38
2011	↑72%	282	↑74%	293	94%	6%	37
2010		164		168	97%	3%	40

* 5 January 2010–20 December 2018. ** in relation to the preceding year.

## Data Availability

Restrictions apply to the availability of these data. Data were obtained from MATES in Construction (Queensland) and are available from the corresponding author with the permission of MATES in Construction.

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
