# Peer review of "The MATES Case Management Model: Presenting Problems and Referral Pathways for a Novel Peer-Led Approach to Addressing Suicide in the Construction Industry"

_ijerph, 2021, doi:10.3390/ijerph18136740_

Round 1

Reviewer 1 Report

The authors have focused on the performance of case management of MATES. Although the topic is interesting, the quality of the conducted research is very poor. So, I would suggest declining this manuscript for publication. The main issues that I decided to recommend for rejecting the manuscript are the lack of adequate justifications for conducting the research as well as the adoption of the poor and unjustified methodology. My comments are as follows:

  • Line 11: MATES should be spelled out.
  • The methodology/methods used in this research should be mentioned in the abstract.
  • Although the problem is stated in this manuscript—which can be stated clearer—the necessity for conducting such research is not justified.
  • The materials and methods section is very poor. There are neither justifications, proper referencing for the chosen methods, nor adequate information related to the analysis.
  • Since the aim of the research and the methodology are not justified, I doubt the validity of the findings.

Reviewer 2 Report

Please see attached comments

Reviewer 3 Report

The study reports data from case management from Mates in Construction, a well-known non-clinical industry-based service provider in the construction industry in Australia. The study design is fairly simple as it is based on routinely collected data and a small exit survey. While I have no major problems with this study, authors could be more critical about the study limitations. The analysis was limited to descriptive statistics. Also, what is known of the reliability of the routinely collected data?

Regarding the Discussion, I wonder if authors could be more clear about the way forward. What are the key lessons and what are the next steps?

Good luck with the revision.

Round 2

Reviewer 1 Report

My main concern is still in the methodology section. I strongly recommend the authors to review similar published research—in top-tier journals—and improve this section significantly. There are so many questions that should be addressed in this section, e.g., why a mixed method is needed? How can you show the robustness of this mixed method? From where you got the questions? what about the reliability of the data and validity of the findings? etc. I cannot accept this manuscript without clear explanations on the Methodology part. My comments are as follows.

Line 102, this is not the aim of the research but the activities/analysis conducted. The authors should re-write this part.

The authors claim that a mixed method is used in this research. Please provide more information on this matter—especially in terms of analysis—, since using open-ended questions will not necessarily prove the employment of a mixed method. The authors may provide a research flowchart at the beginning of “Materials and method” to show the steps taken to achieve the research aim.

Table 1 can be moved to the appendix, and instead, the preparation of the “questions” should be justified in this section.

Reviewer 2 Report

See attached file
